

# Expression profile analysis of cotton fiber secondary cell wall thickening stage

Li Liu[1,*], Corrinne E. Grover[2,*], Xianhui Kong[1], Josef Jareczek[2], Xuwen Wang[1], Aijun Si[1], Juan Wang[1], Yu Yu[1] and Zhiwen Chen[3]

[1] Cotton Institute, Xinjiang Academy of Agricultural and Reclamation Science, Xinjiang, China
[2] Department of Ecology, Evolution and Organismal Biology, Iowa State University, Ames, IA, USA
[3] Engineering Research Center of Coal-based Ecological Carbon Sequestration Technology of the Ministry of Education, Key Laboratory of Graphene Forestry Application of National Forest and Grass Administration, Shanxi Datong University, Datong, China
* These authors contributed equally to this work.

Corresponding authors
Yu Yu, xjyuyu021@sohu.com
Zhiwen Chen, chenzw@yazhoulab.com

## ABSTRACT

To determine the genes associated with the fiber strength trait in cotton, three different cotton cultivars were selected: Sea Island cotton (Xinhai 32, with hyper-long fibers labeled as HL), and upland cotton (17–24, with long fibers labeled as L, and 62–33, with short fibers labeled as S). These cultivars were chosen to assess fiber samples with varying qualities. RNA-seq technology was used to analyze the expression profiles of cotton fibers at the secondary cell wall (SCW) thickening stage (20, 25, and 30 days post-anthesis (DPA)). The results showed that a large number of differentially expressed genes (DEGs) were obtained from the three assessed cotton cultivars at different stages of SCW development. For instance, at 20 DPA, Sea Island cotton (HL) had 6,215 and 5,364 DEGs compared to upland cotton 17–24 (L) and 62–33 (S), respectively. Meanwhile, there were 1,236 DEGs between two upland cotton cultivars, 17–24 (L) and 62–33 (S). Gene Ontology (GO) term enrichment identified 42 functions, including 20 biological processes, 11 cellular components, and 11 molecular functions. Kyoto Encyclopedia of Genes and Genomes (KEGG) enrichment analysis identified several pathways involved in SCW synthesis and thickening, such as glycolysis/gluconeogenesis, galactose metabolism, propanoate metabolism, biosynthesis of unsaturated fatty acids pathway, valine, leucine and isoleucine degradation, fatty acid elongation pathways, and plant hormone signal transduction. Through the identification of shared DEGs, 46 DEGs were found to exhibit considerable expressional differences at different fiber stages from the three cotton cultivars. These shared DEGs have functions including REDOX enzymes, binding proteins, hydrolases (such as GDSL thioesterase), transferases, metalloproteins (cytochromatin-like genes), kinases, carbohydrates, and transcription factors (MYB and WRKY). Therefore, RT-qPCR was performed to verify the expression levels of nine of the 46 identified DEGs, an approach which demonstrated the reliability of RNA-seq data. Our results provided valuable molecular resources for clarifying the cell biology of SCW biosynthesis during fiber development in cotton.

## INTRODUCTION

Cotton is one of the seven major crops and is an essential component of the textile industry with its fiber as the primary product of cotton production (*Chen et al., 2017*; *Huang et al., 2021a*). Cotton is a model plant for studying cellulose synthesis and cell elongation (*Cao et al., 2020b*; *Glover, 2000*; *Guan et al., 2007*). The cotton fiber is a single cell that differentiates from the epidermal cells in the outer integument of the ovule (*Shan et al., 2014*). Cotton fiber development comprises four stages: initiation, elongation, SCW thickening, and maturation (*Huang et al., 2021a*; *Wen et al., 2023*). The fiber elongation stage varies in duration among cultivars and can last between 15 and 25 DPA, depending on the cultivar and its domestication status. Elongation overlaps with the transition stage (16–20 DPA) and, in domesticated cultivars, it also overlaps the beginning of the SCW thickening stage (20 through 40 DPA) (*Mansoor & Paterson, 2012*). Fiber maturity occurs between 40 and 50 DPA, when the fiber cells die and the cytoplasm degrades, leaving behind hollow cells surrounded by cellulose (*Jareczek, Grover & Wendel, 2023*; *Kim, 2018*). The cross-section of the fibers shows the primary cell wall, SCW, luminal wall, and middle luminal wall. During SCW thickening, a helical pattern of cellulose fibers is laid down, resulting in mature fibers appearing flat and banded with a natural twist (*Mansoor & Paterson, 2012*).

Cotton fiber properties are influenced differently during various stages of its growth. The differentiation of cotton fiber from the ovular epidermis happens during initiation (around −1 to 1 DPA) when approximately 20% to 30% of epidermal cells differentiate into fiber cells. During this stage, the fiber tips are refined, which is strongly linked with mature diameter and strength (*Kelly et al., 2015*). The elongation stage, on the other hand, tightly correlates with fiber length. While elongation lasts from ~3 to 20 DPA, 5 to 15 DPA comprises the most rapid elongation period, when cotton leverages fatty acids and carbohydrates to keep the primary cell wall pliable for extreme linear growth (*Tian & Zhang, 2021*). At ~16 DPA, the fiber enters the transition stage, where the microtubules in the fiber shift to a shallow helical angle, and the fiber lays down the winding cell wall layer (*Hsieh, Honik & Hartzell, 1995*; *Meinert & Delmer, 1977*). The winding cell wall layer is similar in composition to the primary cell wall, with a slight increase in cellulose content, and it is thought to impact both fiber strength and flexibility (*Haigler et al., 2009*; *Tuttle et al., 2015*; *Zhang et al., 2021b*). The thickening period of the SCW begins during the transition stage (~15DPA) when cellulose production increases substantially. The SCW synthesis stage starts at ~20 DPA, and it is characterized by β-1, 4-glucan chains that accumulate to facilitate cellulose accumulation and form 20–30 layers of "growing day rings". This period mainly determines the thickness and strength of the cell wall (*Haigler et al., 2012*). When the cell wall thickens to 3–4 microns, the cells begin to dehydrate and undergo apoptosis, and the whole fiber cells display a twisted spiral state

(*Hof & Saha, 1997*). The natural twist of cotton fiber can increase the binding force between fibers and improve the yarn strength when spinning. Mature cotton fibers contain more than 95% cellulose and less keratin, wax, inorganic matter, and other classes of protein than do other types of plant cells (*Liu, 2013*). Cotton fiber quality is based on several properties, such as fiber fineness, length, strength, micronaire (*i.e.*, cell wall thickness), and yellowness (*Bajwa et al., 2015*; *Song et al., 2021*; *Yang et al., 2022*). Increased demand for luxury textiles has likewise increased the demand for high-quality cotton fiber; consequently, there is also increased interest in improving fiber quality in the more highly productive species/cultivars (*Gao et al., 2021*; *Huang et al., 2021a*). While breeding programs are attempting to introgress desirable fiber quality traits into these productive lines, like most crops, cotton fiber yield and quality traits are quantitatively controlled by multiple genes, limiting the success of traditional breeding techniques (*Liu et al., 2023*; *Xu et al., 2019*). Therefore, understanding the physiological and molecular basis of fiber development is paramount to improving cotton fiber quality through other techniques, such as molecular design breeding.

In recent years, the molecular mechanisms underlying cotton fiber development have been studied in depth, and a series of important advances have been made using transcriptome analysis and other high-throughput based methods (*Li et al., 2022*; *Zang et al., 2022*; *Zhang et al., 2022*). These studies, however, mainly focus on the initiation and elongation stages of fiber cell development (*Qin et al., 2019*), and the molecular mechanisms underlying fiber SCW synthesis and thickening (and, therefore, strength) have been rarely studied. To improve our understanding of the molecular mechanisms operating during SCW synthesis and their influence on fiber quality (strength), two cultivars of *Gossypium hirsutum* and one cultivar of *G. barbadense* with known differences in fiber quality were selected for comparative transcriptome analysis, and DEGs from the SCW thickening stage were comprehensively identified using RNA-seq from three timepoints. Through pairwise comparison, common enrichment pathways were identified among the DEGs in the different cultivars, and important candidate genes related to cotton fiber development were screened at different developmental stages. Our results provided a solid foundation for the analysis of the molecular mechanism of cotton fiber SCW development, which would be helpful for the mining and utilization of valuable gene resources.

## MATERIALS AND METHODS

### Plant materials

Cotton cultivars, Xinhai 32 is a high-generation inbred line of Sea Island cotton and 17–24 and 62–33 are high-generation inbred lines of upland cotton. All three cultivars were bred by the Cotton Institute, Xinjiang Academy of Agricultural and Reclamation Science. Xinhai 32, 17–24, and 62–33 are referred to herein as HL, L, and S, respectively. These three cotton cultivars were planted in the field of the Xinjiang Academy of Agricultural and Reclamation Science. Ovules (seeds) were harvested at the indicated days post-anthesis
(DPA). Cotton bolls of 20, 25, and 30 DPA from the three different cotton cultivars were sampled at 10:00 AM. Five cotton bolls were selected from each plant, and the fibers were isolated from the ovules by scratching the ovule with a metal strainer in liquid nitrogen. The fiber samples were then quickly ground into powder and stored in the ultra-low temperature refrigerator at −80 °C.

## Fiber traits and phenotypic evaluation

Boll weight, 100 seed weight, lint index, and seed index were weighed by an analytical balance (0.0001 g, BSA224S, Sartorius, Göttingen, Germany). Fiber quality traits, including the fiber length (mm), fiber uniformity ratio (%), fiber strength (cN/tex), fiber elongation, and micronaire, were measured with an HVI 900 instrument (USTER HVISPECTRUM, SpinLab, Leipzig, Germany) at the Cotton Fiber Quality Inspection and Test Center of Ministry of Agriculture (Anyang, China) (Shang et al., 2015). Bolls were collected for seed index analysis. For this purpose, one hundred seeds from each cultivar were randomly selected and weighed as seed index (SI, g) (Liu et al., 2023; Shang et al., 2016). To measure oil content, seeds were delinted with concentrated sulphuric acid. The oil contents of the three cotton cultivars were measured at different stages of ovular development using the Soxhlet extraction method (García-Ayuso et al., 2000). To record grain weight, one hundred cotton ovules were randomly weighed.

## RNA extraction, library construction, and sequencing

Total RNA from 20, 25, and 30 DPA cotton fiber from each of the three different cotton cultivars was extracted using the RNAprep pure plant kit (Tiangen, Beijing, China) according to the manufacturer's instructions. A total of 1.0 µg purified mRNA was selected for cDNA library construction following a previous report (Chen et al., 2021). Briefly, mRNA was purified from total RNA using poly-T oligo-attached magnetic beads. The cDNA fragments of 240 bp length were selected, and the library fragments were purified using the AMPure XP system (Beckman Coulter, Beverly Hills, CA, USA). The PCR products were purified using the AMPure XP system, and the library quality was assessed on an Agilent Bioanalyzer 2100 system. After generating clusters, the libraries were sequenced using the Illumina HiSeqTM 2500 platform as 150 bp paired-end reads. Three biological replicates were performed for the nine samples.

## RNA-seq reads quality control, mapping, and differentially expressed genes (DEGs) analysis

The FASTX-Toolkit (http://hannonlab.cshl.edu/fastx_toolkit/) was used to process the raw reads in fastq format according to a previous study (Chen et al., 2021). Clean data (clean reads) were obtained by removing reads containing adapters, reads containing poly-N sequences, and low-quality reads from the raw data. All subsequent analyses were performed based on clean high-quality data. Among the nine fiber samples from *G. hirsutum* and *G. barbadense* (Hu et al., 2019; Wang et al., 2019), RNA-seq data were mapped to their respective reference genomes using HISAT2 software

(*Kim, Langmead & Salzberg, 2015*; *Pertea et al., 2016*). Reads with at most one mismatch were used to calculate the expression levels of genes. Gene expression values were calculated following the method of the previous study (*Chen et al., 2022*). Differential expression analysis of the two groups was performed using DESeq2 and presented using fragments per kilobase of transcript per million fragments mapped (FPKM) (*Love, Huber & Anders, 2014*). The resulting $p$ values were adjusted using Benjamini and Hochberg's approach for controlling the false discovery rate (FDR). Genes with an adjusted $p$ value < 0.01 and two-fold change (up and down) were defined as differentially expressed. TBtools (*Chen et al., 2020*) was used to display the gene expression patterns of the FPKM values. Clean data were available from the Genome Sequence Archive in the BIG Data Center of Sciences (https://bigd.big.ac.cn/) under accession number CRA009299. The statistical power of this experimental design, calculated in RNASeqPower is 0.84.

## Gene functional annotation and enrichment analyses

The functions of differentially expressed genes were annotated using the following databases: Nr (NCBI nonredundant protein sequences, ftp://ftp.ncbi.nih.gov/blast/db/), Gene Ontology (GO) (Gene Ontology, http://www.geneontology.org/), and Kyoto Encyclopedia of Genes and Genomes (KEGG) (http://www.genome.jp/kegg/). To analyze the enriched GO of the DEGs, we used the GOseq R package based on Wallenius noncentral hypergeometric distribution (*Young et al., 2010*). KOBAS was employed to assess the statistical enrichment of the DEGs in the KEGG pathways (*Mao et al., 2005*).

## RNA extraction, cDNA synthesis, and RT-qPCR expression analyses

These experiments were conducted according to the methods reported previously (*Cao et al., 2020a*, *2022*; *Cui et al., 2022*; *Zhang et al., 2021a*). In brief, total RNA from 20, 25, and 30 DPA fiber samples of the three different cotton cultivars were extracted using the RNAprep pure plant kit (Tiangen, Beijing, China). DNase I treatment was applied to the RNA samples before synthesizing cDNA using TransScript® First-Strand cDNA Synthesis SuperMix from TransGen Biotech, China, and the resulting products were diluted fivefold before use. Specific forward and reverse gene primers (Table S1) were designed using Primer v5.0 software for real-time quantitative PCR, which was performed using SYBR-Green PCR Mastermix (TaKaRa, Shiga, Japan) on a cycler (Mastercycler RealPlex; Eppendorf Ltd., China). The *G. hirsutum* and *G. barbadense* histone-3 (*GhHIS3* and *GbHIS3*) genes were used as internal references, and the relative amount of amplified product was calculated following the $2^{-\Delta\Delta Ct}$ method (*Livak & Schmittgen, 2001*).

## Statistical analysis

The R package available at https://www.r-project.org/ was utilized for the analysis of variance and Student's t-test. The Shapiro-Wilk test was utilized to test for normality, confirming that the data followed a Gaussian distribution. The least significant difference (LSD) was used to test for significance at either the 1% or 5% levels. The analysis included at least three biological replicates for each sample.

**Table 1 The main properties of different cotton cultivars.**

| Traits | 62–33 (S) | 17–24 (L) | Xinhai 32 (HL) |
|---|---|---|---|
| Boll weight (g) | $5.67 \pm 0.07^a$ | $5.81 \pm 0.07^a$ | $2.69 \pm 0.32^b$ |
| Lint percentage (%) | $39.48 \pm 0.36^a$ | $39.50 \pm 0.21^a$ | $30.44 \pm 0.57^b$ |
| 100 seed weight (g) | $18.50 \pm 0.28^a$ | $19.42 \pm 0.31^a$ | $19.46 \pm 0.28^a$ |
| Lint index (g) | $7.43 \pm 0.21^a$ | $7.50 \pm 0.26^a$ | $6.10 \pm 0.30^a$ |
| Seed index (g) | $11.10 \pm 0.36^a$ | $11.87 \pm 0.06^a$ | $13.33 \pm 0.31^b$ |
| Fiber length (mm) | $28.91 \pm 0.15^a$ | $33.45 \pm 0.44^b$ | $39.84 \pm 0.22^c$ |
| Fiber uniformity ratio (%) | $87.47 \pm 0.56^a$ | $87.63 \pm 0.15^a$ | $90.15 \pm 0.25^b$ |
| Micronaire (MIC) | $4.61 \pm 0.13^a$ | $4.50 \pm 0.14^a$ | $3.83 \pm 0.12^b$ |
| Fiber strength (cN/tex) | $31.57 \pm 1.90^a$ | $37.73 \pm 0.90^b$ | $59.93 \pm 0.90^c$ |
| Fiber elongation (%) | $7.07 \pm 0.06^a$ | $7.03 \pm 0.06^a$ | $7.72 \pm 0.12^b$ |
| Oil content (%) | $28.43 \pm 0.22^a$ | $30.39 \pm 0.18^b$ | $35.35 \pm 0.13^c$ |

Note:
Error bars indicate SD ($n = 3$). Statistically significant differences ("a" is different from "b" or "c", $\alpha = 0.05$ level) of values are indicated with different letters with analysis of variance in R (https://www.r-project.org/). Micronaire value is a comprehensive index reflecting the fineness and maturity of cotton fiber. Micronaire is divided into three levels: A, B and C, with B being the standard level. A grade values range from 3.7 to 4.2 with the best quality; Grade B values range from 3.5 to 3.6 and 4.3 to 4.9. Grade C is below to 3.4 or above to 5.0, showing the worst quality.

# RESULTS

## Physiological traits differences among the *G. barbadense* (Xinhai 32, HL) and *G. hirsutum* (17-24, L and 62-33, S) cotton cultivars

Physiological traits of the three cotton cultivars, all high-generation inbred lines developed by the Cotton Institute, Xinjiang Academy Agricultural and Reclamation Science, were characterized. Table 1 shows the main properties of these cultivars, which we have designated as S (cv 62–33), L (cv 17–24), and HL (Xinhai 32). In comparison to the upland cotton cultivars (L and S), the Sea Island cotton (HL) exhibited longer fiber, greater fiber strength, increased oil content, a higher seed index, greater fiber uniformity, and greater fiber elongation. In contrast, the upland cotton cultivars (L and S) had significantly higher boll weight, lint percentage, lint index, and micronaire than did the HL cultivar. Notably, these properties, although greater in the *G. hirsutum* cultivars, are outside of the optimal range. In particular, micronaire (one of the most important measures of cotton fiber quality) was considered grade A (micronaire range: 3.7–4.2) in HL, compared the B grade (micronaire range: 3.5–3.6 and 4.3–4.9) observed in the upland cotton cultivars, L and S. Between the two cultivars of upland cotton, fiber length, fiber strength, and oil content also varied, with cv 17–24 (L) exhibiting significantly higher values than cv 62–33 (S). These results indicate that major differences in agronomically important fiber properties exist between Sea Island cotton and upland cotton, as expected, but also between the two upland cotton cultivars, with cv 17–24 (L) exhibiting better agricultural performance.

## Transcriptome data generation of 20, 25, and 30 DPA fibers of Xinhai 32 (HL), 17–24 (L), and 62–33 (S) cotton cultivars

As these three cultivars showed great variance in fiber length and strength, especially fiber strength (Table 1), we evaluated gene expression at three timepoints during SCW synthesis

**Table 2 Mapping results of RNA-seq clean reads from nine fiber samples.**

| Samples | Map to genome | 20 DPA Number | Percentage | 25 DPA Number | Percentage | 30 DPA Number | Percentage |
|---|---|---|---|---|---|---|---|
| HL | Total reads | 12.1 M | 100% | 12.3 M | 100% | 11.6 M | 100% |
| | Total base pairs | 590.9 Mbp | 100% | 600.8 Mbp | 100% | 569.0 Mbp | 100% |
| | Total mapped reads | 9.3 M | 77% | 9.3 M | 76% | 8.7 M | 75% |
| | Perfect match | 5.2 M | 43% | 5.1 M | 41.86% | 4.8 M | 40.99% |
| | <=2 bp mismatch | 4.1 M | 34% | 4.2 M | 34% | 4.0 M | 34% |
| | Unique match | 8.1 M | 67% | 8.4 M | 68% | 7.7 M | 65.96% |
| | Multi-position match | 1.2 M | 10% | 0.97 M | 7.91% | 1.1 M | 9% |
| | Total unmapped reads | 2.8 M | 23% | 2.9 M | 23.82% | 2.9 M | 24.82% |
| L | Total reads | 12.1 M | 100% | 11.7 M | 100% | 11.9 M | 100% |
| | Total base pairs | 593.0 Mbp | 100% | 573.5 Mbp | 100% | 583.3 Mbp | 100% |
| | Total mapped reads | 9.2 M | 76% | 8.9 M | 76% | 8.5 M | 72% |
| | Perfect match | 5.0 M | 41% | 4.9 M | 42% | 4.9 M | 41% |
| | <=2 bp mismatch | 4.1 M | 34% | 4.0 M | 34% | 3.6 M | 30% |
| | Unique match | 8.5 M | 70% | 8.1 M | 69% | 6.3 M | 53% |
| | Multi-position match | 0.68 M | 6% | 0.79 M | 7% | 2.2 M | 19% |
| | Total unmapped reads | 2.9 M | 24% | 2.8 M | 24% | 3.4 M | 28% |
| S | Total reads | 12.6 M | 100% | 11.7 M | 100% | 12.2 M | 100% |
| | Total base pairs | 616.5 Mbp | 100% | 573.5 Mbp | 100% | 598.9 Mbp | 100% |
| | Total mapped reads | 9.4 M | 75% | 8.9 M | 76% | 9.1 M | 75% |
| | Perfect match | 5.2 M | 41% | 4.9 M | 42% | 5.0 M | 41% |
| | <=2bp mismatch | 4.2 M | 34% | 4.0 M | 34% | 4.2 M | 34% |
| | Unique match | 8.7 M | 69% | 8.2 M | 70% | 8.3 M | 68% |
| | Multi-position match | 0.69 M | 5% | 0.72 M | 6% | 0.80 M | 7% |
| | Total unmapped reads | 3.2 M | 25% | 2.8 M | 24% | 3.1 M | 25% |

(*i.e.*, 20, 25, and 30 DPA fiber) from Xinhai 32 (HL), 17–24 (L), and 62–33 (S). RNA from at least three biological replicates was pooled at each timepoint for each accession, hereafter referred to as HL20, HL25, HL30, L20, L25, L30, S20, S25, and S30, and over 108 million high-quality reads were generated using the Illumina HiSeqTM 2500 sequencing platform (Table 2). The HISAT2 software was used to align these clean reads to the reference cotton genomes (*Hu et al., 2019*; *Wang et al., 2019*) with at most one base mismatch. The ratio of mapped reads ranged from 72% in sample L30 to 77% in sample HL20 (Table 2), and the number of uniquely mapped clean reads ranged from 53% in sample L30 to 70% in sample L20. These data indicated that the RNA-seq data in this study were reliable for the subsequent analyses.

## Identification of differentially expressed genes

Differential gene expression was surveyed for the nine cotton fiber samples. Eighteen comparisons among these nine samples were performed to capture expression differences between samples at an assessed timepoint and within a single sample at each timepoint

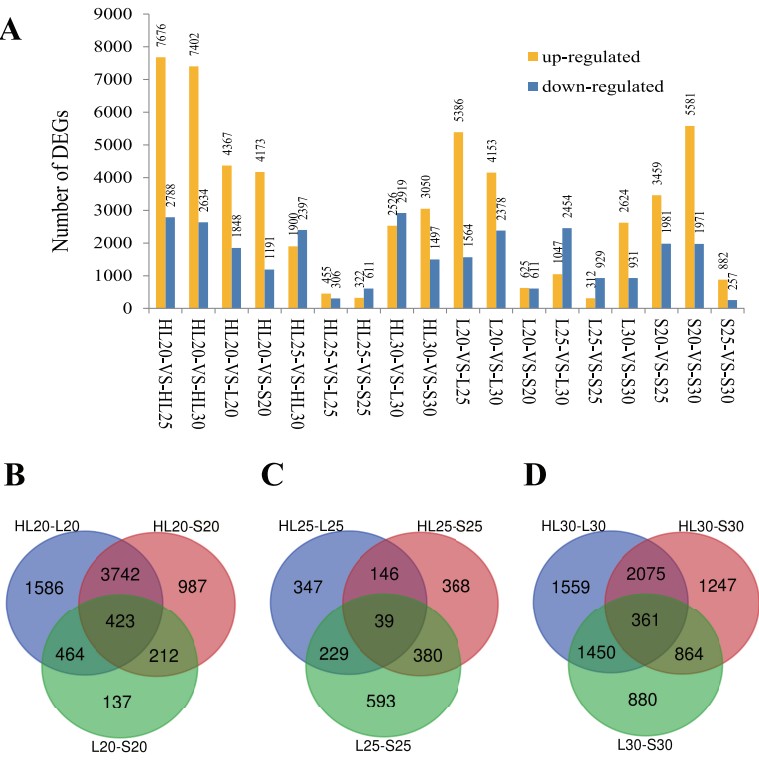

**Figure 1** **Differentially expressed genes (DEGs) identified among three different cotton cultivars at 20, 25, and 30 DPA fiber samples.** (A) DEGs identified among 18 paired comparisons, (B) Venn diagram comparisons of DEGs among cultivars at 20 DPA, (C) Venn diagram comparisons of DEGs among cultivars at 25 DPA, (D) Venn diagram comparisons of DEGs among cultivars at 30 DPA. Note: Comparison of sample A with B was designated as A-*VS*-B, in which A was the control and B was the treatment. HL, L, and S represent accessions and 20, 25, and 30 represent the days post anthesis (DPA).

(Fig. 1A). In each comparison, the number of differentially expressed genes (DEGs) varied between 761 genes in HL25-*VS*-L25 and 10,464 genes in HL20-*VS*-HL25 (Fig. 1A). It is worth noting that upregulation was more frequent than downregulation in most comparisons (Fig. 1A). Because these cultivars produce fibers with different properties, we considered the overlap in gene expression at each assessed timepoint to identify genes important for SCW biosynthesis that differed in expression among the three species/cultivars. Venn diagrams were constructed for the DEGs from each species/cultivar comparison at each surveyed stage (*i.e.*, 20, 25, and 30 DPA stages; Fig. 1). As expected, the intraspecies cultivar comparisons (*i.e.*, L *vs* S) typically exhibited fewer uniquely DEGs than the interspecies comparisons (*i.e.*, HL *vs* S or L), with the exception of the 25 DPA stage of development (Fig. 1). Although at the 25 DPA timepoint fewer DEGs were identified (Fig. 1), the intraspecies (S *vs* L) comparison resulted in 25–40% more DEGs than either interspecific comparison (Fig. 1); the fewest DEGs at this stage were between the HL and L cultivars, perhaps indicating that underlying fiber length expression differences are largely responsible for the differences in gene expression at this stage. The results showed that 423 DEGs were shared by all species/cultivar comparisons at the 20 DPA stage (Fig. 1B), possibly indicating genes that underlie the differences among cultivars. Far fewer DEGs

were observed at 25 DPA (39 DEGs), although this number is commensurate with the general reduction in DEGs among cultivars at this stage (Fig. 1C). At 30 DPA, 361 DEGs were shared among the cultivar comparisons (Fig. 1D).

## GO analysis for DEGs

We considered the possible biological functions of these DEGs between Sea Island and/or the upland cottons using GO category enrichment (Fig. S1). Results from the three categories, including biological process, cellular component, and molecular function, suggest enrichment of 20, 11, and 11 functional categories in the 20, 25, and 30 DPA fiber comparisons, respectively. GO terms associated with important biological processes included metabolic, cellular, developmental, and single-organism processes, biological regulation, response to stimulus, and signaling. Cellular components, such as cell, cell part, membrane, membrane part, organelle and organelle parts were enriched. Molecular function enrichment consisted of catalytic activity, transporter activity, binding, nucleic acid-binding transcription factor activity, antioxidant activity, and receptor activity.

To compare the difference between the three cultivars, the GO enrichment of functional categories in the 25 DPA fiber was analyzed (Fig. 2). The results showed that the top three GO category enrichments in biological processes among the three groups of comparisons (HL-25-*vs*-L-25, HL-25-*vs*-S-25, L-25-*vs*-S-25) were cellular process, metabolic process, and localization, respectively (Figs. 2A–2C). Additionally, the cellular components, cell, cell part and membrane accounted for the largest proportion in all three groups (Figs. 2A–2C). The most enriched molecular functions all consisted of catalytic activity, and transporter activity between the comparisons of the three cultivars (Figs. 2A–2C). These results indicated that the GO enrichment analysis was not specific to the three cultivars.

## Pathway enrichment of DEGs from the three cotton cultivars at different stages of fiber development (20, 25, and 30 DPA)

To investigate the biological functions of these DEGs during fiber development in the three cotton cultivars, we performed KEGG pathway enrichment analysis for the DEGs. At the 20 DPA fiber stage, the DEGs were assigned to 126 KEGG pathways according to the functional categorization. For the interspecific comparison HL-20-*vs*-L-20, the top 20 KEGG pathways of enriched DEGs were categorized into the following functional pathways (Fig. 3A). While the greatest number of DEGs mostly belonged to the plant hormone signal transduction pathways, there was also involvement of several other pathways whose RichFactor was closer to 50%, such as fatty acid elongation. Other pathways with notable enrichment are: i) carbohydrate metabolism: glycolysis/ gluconeogenesis, pyruvate metabolism, galactose metabolism, fructose and mannose metabolism; ii) fatty acid metabolism: fatty acid elongation, and biosynthesis of unsaturated fatty acids; and iii) amino acid metabolism: valine, leucine and isoleucine degradation, lysine biosynthesis, and cysteine and methionine metabolism. For the other interspecific comparison, HL-20-*vs*-S-20, the top 20 KEGG pathways of enriched DEGs were involved in glycolysis/gluconeogenesis, galactose metabolism, fructose and mannose metabolism, fatty acid elongation, biosynthesis of unsaturated fatty acids, and leucine and

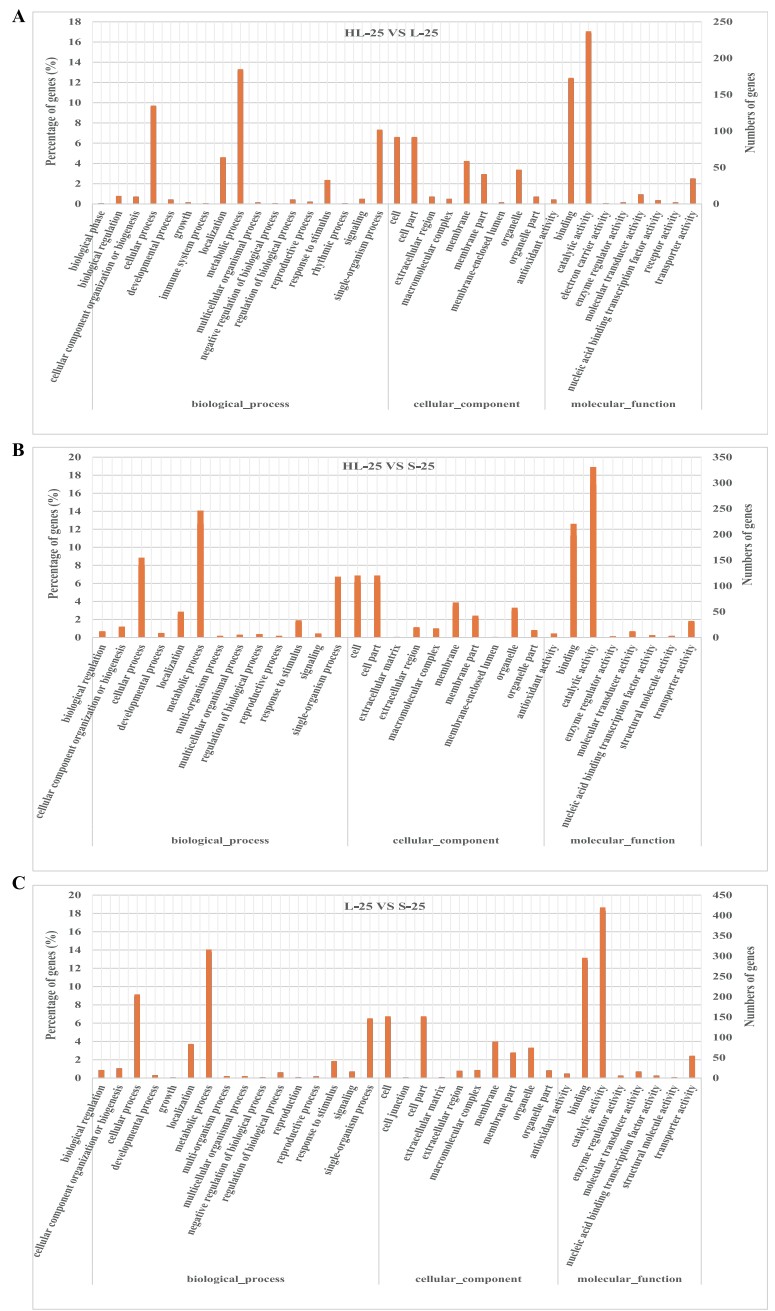

**Figure 2 Gene ontology (GO) enrichment analysis of DEGs in 25 DPA fiber samples between Sea Island or upland cottons.** (A) GO enrichment analysis between HL-25-*vs*-L-25, (B) GO enrichment analysis between HL-25-*vs*-S-25, (C) GO enrichment analysis between L-25-*vs*-S-25. The X-axis represents the biological functions (molecular function, biological process, and cellular component) of these DEGs. The Y-axis represents the percentage or number of genes categorized into different functional pathways.

isoleucine degradation (Fig. 3B); however, the greatest representation of genes was for RNA transport and the ribosome. In this comparison, the RichFactor varied more narrowly (~12%, *vs* 25% in HL20 *vs* L20). For the intraspecific comparison L-20-*vs*-S-20,

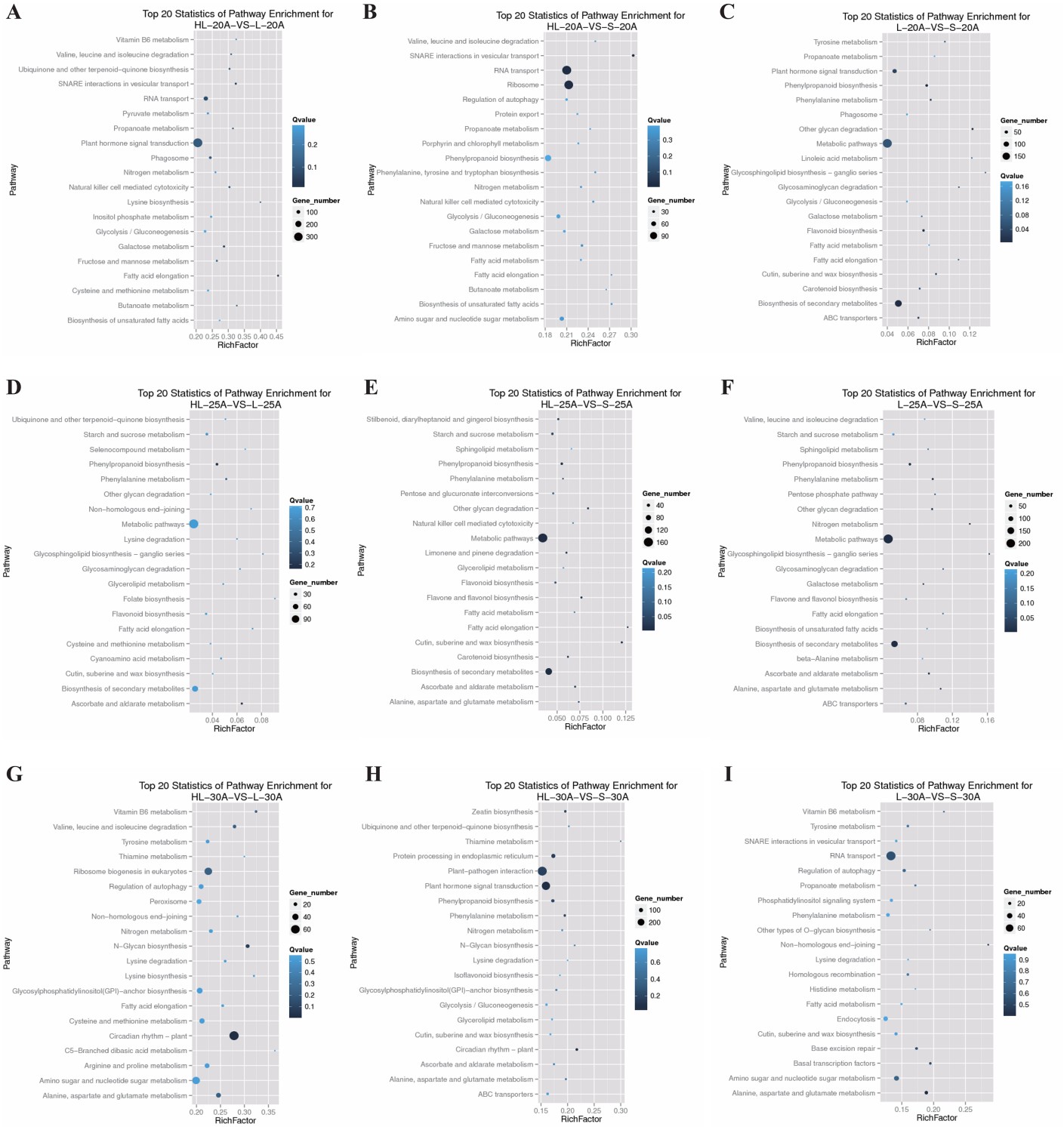

**Figure 3 KEGG pathway analysis of enriched differentially expressed genes.** The "RichFactor" (x-axis) represents the ratio of differentially expression genes *vs* all genes in that pathway. Circle sizes correspond to gene number, and the q-value is given for each analysis. (A) Top 20 pathways of significantly enriched DEGs from HL-20 *vs* L-20, (B) top 20 pathways of significantly enriched DEGs from HL-20 *vs* S-20, (C) top 20 pathways of significantly enriched DEGs from L-20 *vs* S-20, (D) top 20 pathways of significantly enriched DEGs from HL-25 *vs* L-25, (E) top 20 pathways of significantly enriched DEGs from HL-25 *vs* S-25, (F) top 20 pathways of significantly enriched DEGs from L-25 *vs* S-25, (G) top 20 pathways of significantly enriched DEGs from HL-30 *vs* L-30, (H) top 20 pathways of significantly enriched DEGs from HL-30 *vs* S-30, (I) top 20 pathways of significantly enriched DEGs from L-30 *vs* S-30.            

the top 20 KEGG pathways of enriched DEGs mainly included the glycolysis/gluconeogenesis, galactose metabolism, fatty acid elongation, biosynthesis of unsaturated fatty acids, plant hormone signal transduction, and biosynthesis of secondary metabolites (Fig. 3C). The greatest numbers of genes were found in the metabolic pathways and biosynthesis of secondary metabolites. While the RichFactor for these was low, it is worth noting that the RichFactor was generally low for all pathways, possibly indicating the recruitment of few genes/pathways from these somewhat broad categories. These results suggested that the differences in the 20 DPA fiber samples between Sea Island cotton (Xinhai 32) and upland cotton (17–24 or 62–33) were mainly caused by genes involved in the metabolic pathways of carbohydrates, fatty acids, and amino acids. However, differences in the plant hormone pathways were common to both comparisons involving L-20, and differences in secondary metabolite production were observed between the two cultivars of upland cotton.

Inter-cultivar comparisons revealed that the 25 DPA fiber stage mainly exhibited enriched KEGG pathways for DEGs related to starch and sucrose metabolism, fatty acid elongation, and biosynthesis of secondary metabolites (Figs. 3D–3F), which could indicate that secondary metabolites, such as the biosynthesis of cellulose, was different among the three cultivars. For the fiber samples of 30 DPA, the enrichment classification of DEGs among the three cultivars did not show a high degree of similarity, indicating the complexity of the genotypes among different cotton cultivars (Figs. 3G–3I).

## Annotations of DEGs from different fiber development stages

To eliminate the effect of genotypic differences, KEGG pathway enrichment was performed on DEGs between different stages of fiber development (20, 25, and 30 DPA) within the same cotton species or cultivar (Fig. 4). The shared top 20 KEGG pathways of enriched DEGs from Xinhai 32 (HL) were mainly categorized into the following functional pathways: pyruvate metabolism, fatty acid elongation, biosynthesis of unsaturated fatty acids, valine, leucine and isoleucine degradation, as well as plant hormone signal transduction (Figs. 4A–4C). As to the upland cotton 17–24 (L) (Figs. 4D–4F) and 62–33 (S) (Figs. 4G–4I), the enriched DEGs were also mainly categorized into the fatty acid elongation, biosynthesis of unsaturated fatty acids, valine, leucine and isoleucine degradation, circadian rhythm as well as plant hormone signal transduction pathways. Consistent with the previous results, these results further supported that DEGs enriched in the pathways of carbohydrates, fatty acids, amino acids, and hormones were involved in the elongation of fiber cells or thickening of fiber SCW of different cotton species at 20–30 DPA stages.

## Identification of shared DEGs from different fiber development stages in the three cotton cultivars

To further identify candidate genes contributing to the biosynthesis and thickening of cotton fiber SCW, the shared DEGs from different fiber development stages in the three cotton cultivars were evaluated. As can be seen from Fig. S2, there were 46 DEGs shared by the three cotton cultivars, Xinhai 32 (HL), 17–24 (L), and 62–33 (S) at different fiber

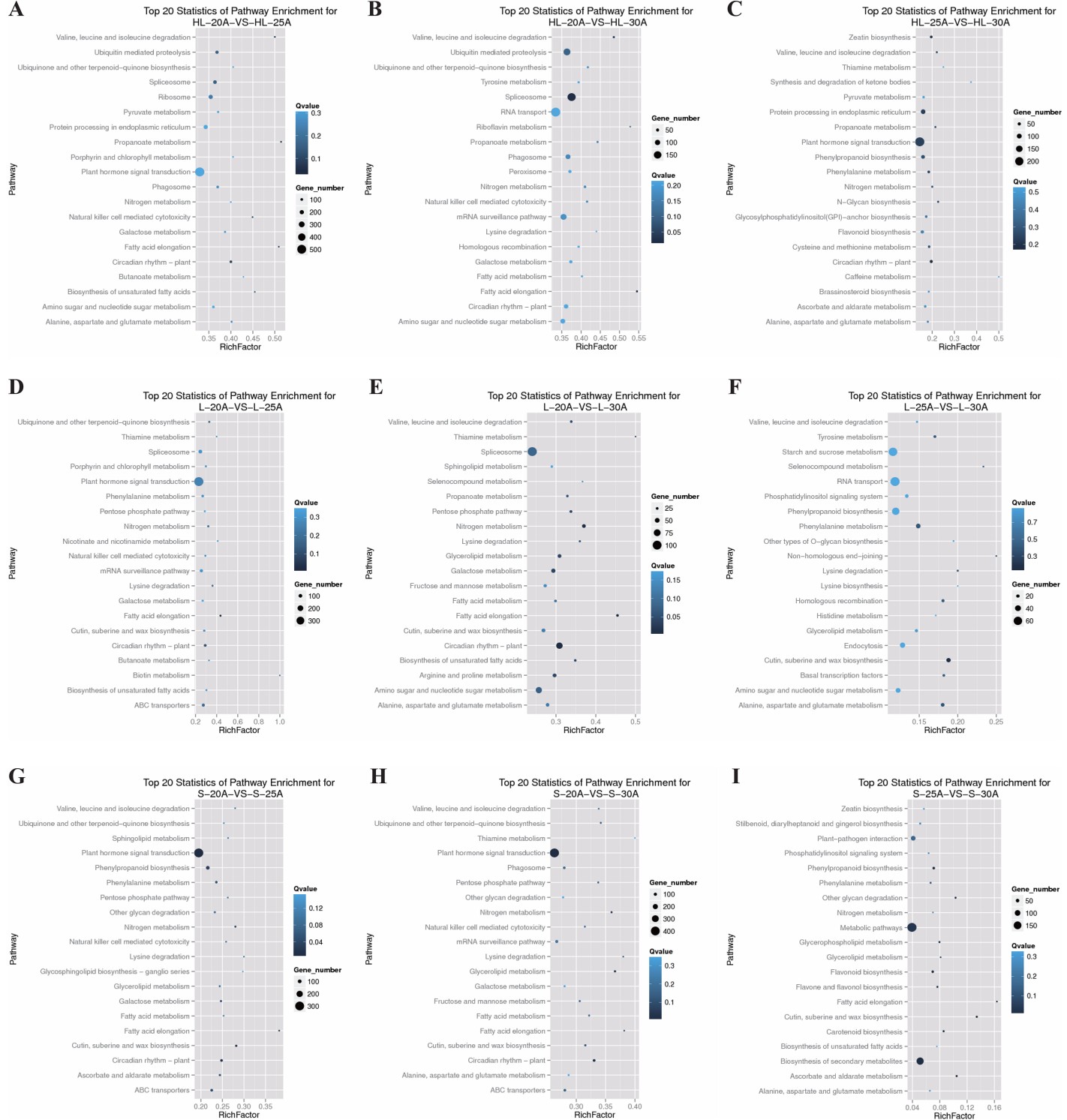

**Figure 4 KEGG pathway analysis of enriched differentially expressed genes.** (A) Top 20 pathways of significantly enriched DEGs from HL-20 *vs* HL-25, (B) top 20 pathways of significantly enriched DEGs from HL-20 *vs* HL-30, (C) top 20 pathways of significantly enriched DEGs from HL-25 *vs* HL-30, (D) top 20 pathways of significantly enriched DEGs from L-20 *vs* L-25, (E) top 20 pathways of significantly enriched DEGs from L-20 *vs* L-30, (F) top 20 pathways of significantly enriched DEGs from L-25 *vs* L-30, (G) top 20 pathways of significantly enriched DEGs from S-20 *vs* S-25, (H) top 20 pathways of significantly enriched DEGs from S-20 *vs* S-30, (I) top 20 pathways of significantly enriched DEGs from S-25 *vs* S-30.

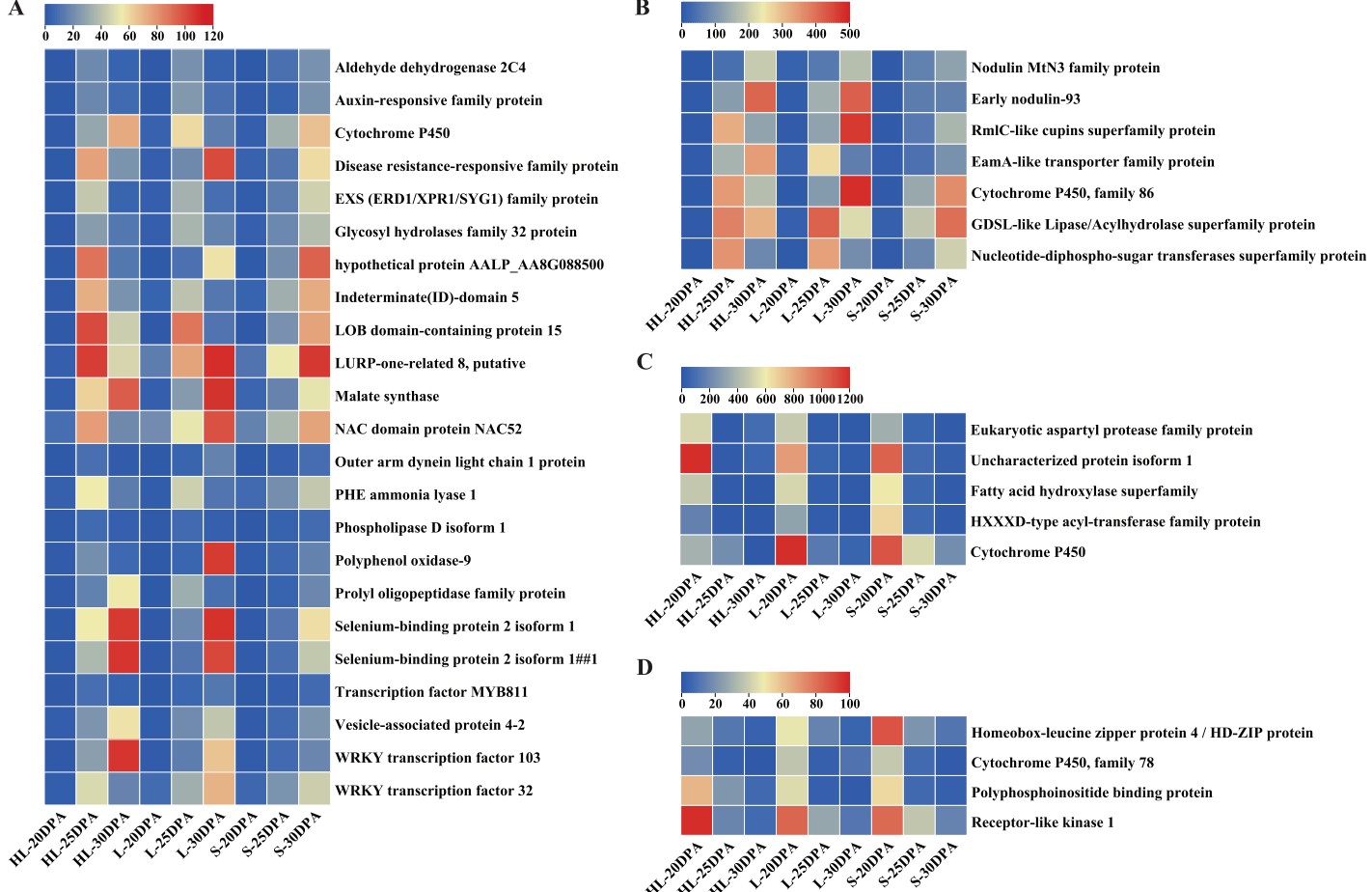

**Figure 5 Expression patterns of DEGs across the nine fiber samples.** (A) 23 DEGs with low expression at 20 DPA but high expression at 25 or 30 DPA (*i.e.*, low-to-high expression), (B) seven additional DEGs with low-to-high expression, (C) five DEGs with high expression at 20 DPA but low expression at 25 or 30 DPA (*i.e.*, high-to-low expression), (D) four additional DEGs showing high-to-low expression. Note: HL represents Xinhai 32 (Sea Island cotton), L represents 17–24 (upland cotton), and S represents 62–33 (upland cotton). DPA, days post anthesis.

development stages (20, 25, and 30 DPA). These 46 recurring DEGs may be common to the fiber developmental pathway, regardless of the cultivar, perhaps suggesting they could be key genes in the regulation of cotton fiber cell elongation or SCW thickening. Their main functions include REDOX enzymes, selenium and polyphosphoinositide binding proteins, hydrolases (such as GDSL thioesterase), transferases, metalloproteins (cytochromatin-like genes), kinases, carbohydrates, and transcription factors (MYB and WRKY).

Detailed analyses were conducted on these DEGs to assess their expression patterns across the three assessed timepoints of the three cultivars (Fig. 5). In general, the DEGs can be classified into two categories (Fig. 5): low-to-high expression and high-to-low expression. Most of the DEGs (30) showed low-to-high expression, starting relatively low at 20 DPA but increasing in expression at 25 or 30 DPA in each of the three different species or cultivars (Figs. 5A, 5B). Importantly, the expression levels of these 30 DEGs in HL or L were significantly higher than that in S at 25 or 30 DPA (Figs. 5A, 5B), suggesting

that these genes might play important roles in the SCW synthesis and thickening of fiber tissues. In contrast, nine of the remaining DEGs exhibited high-to-low expression trends, exhibiting the greatest expression at 20 DPA but reducing expression by 25 or 30 DPA (Figs. 5C, 5D). These results suggest that these genes may function in the early stages of fiber development and are downregulated as the cell commits to focused SCW synthesis.

## Validation of candidate DEGs by RT-qPCR

We have confirmed the accuracy of these candidate gene expression profiles for nine of the differentially expressed genes (*i.e.*, Polyphenol oxidase 9, 3-ketoacyl-CoA synthase 3, NAD (P)-binding Rossmann-fold superfamily protein, GDSL-like lipase/acylhydrolase superfamily protein, caffeic acid O-methyltransferase 1, WRKY transcription factor 32, 2,4-dienoyl-CoA reductase, WRKY transcription factor 103 and fatty acid desaturase 6) using RT-qPCR (Fig. 6). The results showed that eight candidate genes have low-to-high expression trends, presenting high levels of expression at 25 or 30 DPA compared with 20 DPA (Figs. 6A–6H). One candidate gene (3-ketoacyl-CoA synthase 3) exhibited a high-to-low expression pattern with high expression levels at 20 DPA but low expression levels at 25 or 30 DPA (Fig. 6I). Overall, these transcripts of nine genes exhibited similar expression patterns between the RT-qPCR and RNA-seq experiments (Fig. S3), and according to the results of correlation analysis (Table S2), the correlation coefficient between the RT-qPCR and RNA-seq data of nine genes ranged from 0.8562 to 0.9997. These results indicated that the RT-qPCR validation was highly consistent with the data of the initial RNA-seq analysis. Both data proved that the expression profiles of 2,4-dienoyl-CoA reductase, WRKY103, and fatty acid desaturase 6 genes in HL or L were significantly higher than that in S at 25 or 30 DPA (Figs. 6F–6H), suggesting that the diverged expression patterns of these genes may be the cause of the variance in fiber strength between the three cultivars.

## DISCUSSION

Cotton fiber cell development is a complex morphogenic process regulated by the tightly controlled timing of the expression of multiple genes. Early research suggested that the timing of different fiber developmental stages lacks similarity between Sea Island and upland cotton species, specifically in the extent to which the elongation stage of fiber cell development overlaps with the SCW thickening stage (*Zang et al., 2022*; *Zhang et al., 2022*). In this study, we evaluated 20, 25, and 30 DPA fibers from Sea Island and upland cotton cultivars, representing developmental stages of the SCW biosynthesis or thickening. It is worth noting that pathways related to fatty acid metabolism were highly enriched, such as the fatty acid carbon chain extension pathway, propionic acid metabolic pathway, and unsaturated fatty acid biosynthesis pathway. Their differential expression during the synthesis process of the fiber SCW suggested that fatty acid metabolism was closely related to fiber quality.

Further, a total of 46 genes were screened as candidates that were commonly differentially expressed during the different development stages from the three cotton cultivars, including cytochrome P450 enzyme gene, glycohydrolase and glycosyltransferase, selenium and polyphosphoinositide binding protein genes, WRKY

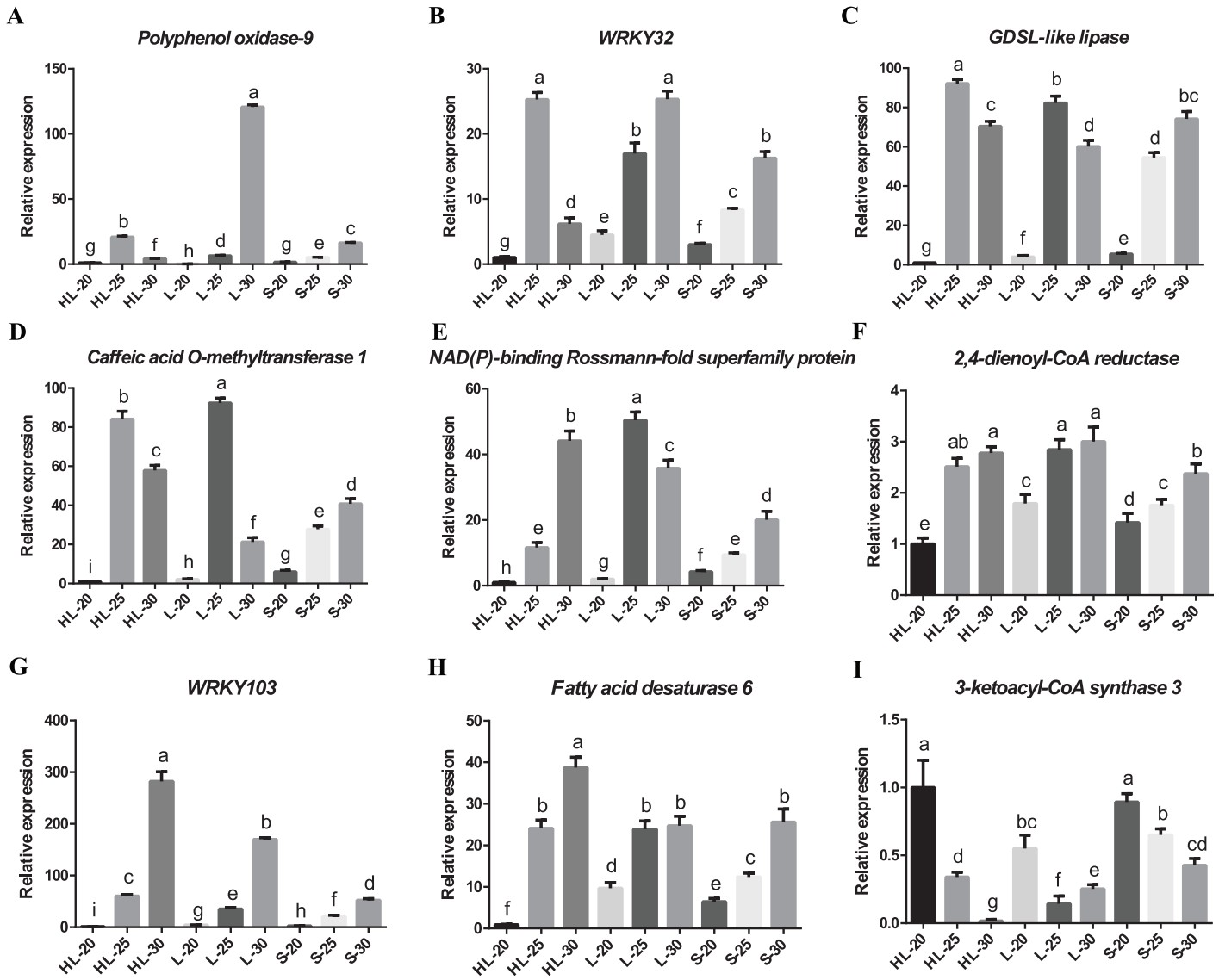

**Figure 6   Real-time quantitative PCR validation of DEGs from RNA-seq data.** Relative expression of (A) *Polyphenol oxidase-9*, (B) *WRKY32*, (C) *GDSL-like lipase*, (D) *Caffeic acid* O-*methyltransferase 1*, (E) *NAD(P)-binding Rossmann-fold superfamily protein*, (F) *2,4-dienoyl-CoA reductase*, (G) *WRKY103*, (H) *Fatty acid desaturase 6*, (I) *3-ketoacyl-CoA synthase 3* genes at 20, 25, or 30 DPA fiber cells of the three cotton cultivars, and the expression level in the HL-20 sample was set to 1 (means of triplicates ± SD). Note: HL represents Xinhai 32 (Sea Island cotton), L represents 17–24 (upland cotton), and S represents 62–33 (upland cotton). Relative gene expression levels are normalized to histone-3 gene values. Error bars indicate SD ($n = 3$). Statistically significant differences ("a" is different from "b", "c", "d", "e" "f", "g", "h", or "i", α = 0.05 level) of expression values are indicated with different letters with analysis of variance in R (https://www.r-project.org/).                   

transcription factor (Fig. 5). Plant cytochrome P450s are involved in the biosynthetic pathways of fatty acid hydroxylation, epoxidation and cleavage of hydrogen peroxide functional groups of unsaturated fatty acids (*Davidson, Reid & Helliwell, 2006*). Previous reports have detailed that ethylene biosynthesis, cytoskeleton, signaling pathway, fatty acid biosynthesis and fatty acid carbon chain extension pathway (*Shi et al., 2006*) were significantly up-regulated during fiber development (*Gou et al., 2007*; *Ruan et al., 2004*).

These results indicated that lipid metabolism was significantly correlated with fiber development.

It is well-known that MYB transcription factors promote secondary cell-wall biosynthesis (*Xiao et al., 2021*). GhMYBL1, an R2R3-MYB transcription factor, was specifically expressed at the stage of SCW deposition in cotton fibers and is involved in modulating the process of SCW biosynthesis (*Sun et al., 2015*). Additionally, GhMYB7 has been shown to regulate the biosynthesis of SCWs both in *Arabidopsis thaliana* and upland cotton (*Huang et al., 2021b*, *2016*). In our study, a MYB transcription factor was identified in the highly expressed genes of the three cultivars at 25 and 30 DPA fiber, and may be involved in the SCW cellulose biosynthesis. WRKY members have also been reported to be involved in fiber development, such as GhWRKY16, and GhWRKY53 (*Wang et al., 2021*; *Yang et al., 2021*). Two WRKY genes (WRKY32 and 103) were also detected in our data, and their potential biological roles in regulating SCW biosynthesis of fiber cells were still not clear, which is worth further research to explore their functions. Thus, we speculated that one factor that cause the difference in fiber strength of the three cultivars might be the distinct expression patterns of genes related to the SCW biosynthesis.

Many other biologically relevant genes were also identified to be specifically differentially expressed in cotton fibers, such as the cotton sucrose synthetase gene (*Zhang et al., 2017*), transcription factor GhMYB2 (*Wang et al., 2004*), cytoskeletal proteins GhTUB1 and GhACT1, which have been shown to participate in the elongation process of fiber cells (*Li et al., 2005*). GhACT1 and GhTUB1, which encode actin and tubulin, also affect cytoskeleton assembly and fiber elongation (*Li et al., 2002*, *2005*). LIM domain protein GhWLIM1a can promote SCW synthesis by binding to tubulin (*Han et al., 2013*). Transcription factors were also involved in regulating the SCW synthesis of cotton fiber cells. Overexpression of a cotton NAC transcription factor (GhFSN1) resulted in thicker fiber SCWs but shorter fibers (*Zhang et al., 2018*). Subsequently, a primary GhTCP4 transcription factor was found to play an important role in balancing cotton fiber cell elongation and SCW thickening (*Cao et al., 2020b*). Transcriptomic and promoter activity analysis showed that GhTCP4 activated GhFSN1 transcription factor and cellulose synthase genes responsible for SCW synthesis. The transcriptional activity of GhCESA8 (GhCESA8) accelerated the biosynthetic pathway of the SCWs of fiber cells, resulting in shorter fibers and thicker cell walls (*Cao et al., 2020b*). The time-course analysis of fiber samples in *G. hirsutum* and *G. barbadense* cultivars revealed that the glycosyltransferase was involved in the synthesis of glucuronoxylan hemicellulose and cell wall morphogenesis during SCW formation (*Zhang et al., 2022*). A recent study on the genetic regulation of fiber development in *G. hirsutum* based on 2,215 time-series transcriptomes also revealed that a NAD(P)-linked oxidoreductase protein, was positively correlated with fiber strength development (*You et al., 2023*). In our study, the NAC52 transcription factor, and NAD(P)-binding Rossmann-fold superfamily protein, have been identified to be involved in the regulation of cotton fiber SCW development. These results indicated that multiple pathway-related genes play roles in the biosynthesis of fiber SCWs.

## ABBREVIATIONS

| | |
|---|---|
| **DEGs:** | differentially expressed genes |
| **FPKM:** | fragments per kilobase of transcript per million mapped fragments |
| ***G. hirsutum*:** | *Gossypium hirsutum* |
| ***G. barbadense*:** | *Gossypium barbadense* |
| ***A. thaliana*:** | *Arabidopsis thaliana* |
| **RT-qPCR:** | real-time quantitative polymerase chain reaction |
| **GO:** | Gene Ontology |
| **KEGG:** | Kyoto Encyclopedia of Genes and Genomes |
| **SCW:** | secondary cell wall |

## ACKNOWLEDGEMENTS

We are deeply indebted to Professor Lida Zhang for helpful suggestions and comments on bioinformatic analyses, and we thank Professor Yi Huang for valuable comments on previous versions of the manuscript. We are also grateful to two anonymous reviewers for their helpful suggestions and comments.

### Funding

The work reported in this publication was supported by the National Key Research and Development Program of China (No. 2023YFD2301200), the National Natural Science Foundation of China (31360349). The funders had no role in study design, data collection and analysis, decision to publish, or preparation of the manuscript.

### Grant Disclosures

The following grant information was disclosed by the authors:
National Key Research and Development Program of China: 2023YFD2301200.
National Natural Science Foundation of China: 31360349.

### Competing Interests

The authors declare that they have no competing interests.

### Author Contributions

- Li Liu conceived and designed the experiments, performed the experiments, analyzed the data, prepared figures and/or tables, authored or reviewed drafts of the article, and approved the final draft.
- Corrinne E. Grover analyzed the data, prepared figures and/or tables, authored or reviewed drafts of the article, and approved the final draft.
- Xianhui Kong performed the experiments, prepared figures and/or tables, and approved the final draft.
- Josef Jareczek analyzed the data, prepared figures and/or tables, authored or reviewed drafts of the article, and approved the final draft.

- Xuwen Wang performed the experiments, prepared figures and/or tables, and approved the final draft.
- Aijun Si performed the experiments, prepared figures and/or tables, and approved the final draft.
- Juan Wang performed the experiments, authored or reviewed drafts of the article, and approved the final draft.
- Yu Yu conceived and designed the experiments, authored or reviewed drafts of the article, and approved the final draft.
- Zhiwen Chen conceived and designed the experiments, analyzed the data, prepared figures and/or tables, authored or reviewed drafts of the article, and approved the final draft.

### Data Availability

The RNA-sequencing data are available at the Genome Sequence Archive in the BIG Data Center of Sciences: CRA009299.

### Supplemental Information

Supplemental information for this article can be found online at http://dx.doi.org/10.7717/peerj.17682#supplemental-information.

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
