# Peer review of "Expression profile analysis of cotton fiber secondary cell wall thickening stage"

_PeerJ, doi:10.7717/peerj.17682_

## Round 0.1 · original submission · Major Revisions

Please carefully consider all concerns raised by Reviewer #1 post assessment of your study.

I have also assessed your manuscript and I share all the concerns raised by Reviewer #1. Therefore, these concerns must be addressed via the production of a significantly revised manuscript version.

A significant concern is the lack of evidence that the DEGs identified play any biologically relevant role in secondary cell wall synthesis as part of cotton fiber development. Such evidence must be provided in the revised manuscript version.

Please also improve the degree of impact/amount of information being presented in each of the Figures of your study. Specifically:

- Figures 1 and 2 do not carry enough weight to be standalone Figures. Please combine these two Figures in the revised manuscript to improve the impact of the data being presented.

- Figure 3 also provides little impact - the addition of further information would be beneficial to the reader and would improve Figure impact.

- Figure 6 does not present enough data to form a standalone Figure. Please add additional data to improve the standard / impact of this Figure.

- RT-qPCR data presented in Figure 8 should be presented as a bar graph so that statistical analyses can be included. Current presentation of the RT-qPCR data in Figure 8 is not of an acceptable standard.


Reviewer 1 ·

Basic reporting

The mature cotton fiber has a unique cell wall structure that is distinct from xylem of monocot and dicot plants due to the massive deposition of cellulose during the fiber SCW thickening stage. Cotton fiber cell walls not only determine fiber morphogenesis, but also determine fiber quality such as length and strength. It is therefore of interest to understand SCW biosynthesis and regulation for genetic improvement of cotton fibers. Liu et al selected three cotton cultivars: Sea Island cotton (Xinhai 32) and upland cotton (17-24 and 62-33) which have fibers with different qualities and performed RNA-seq analysis in order to identify candidate genes involved in fiber SCW thickening.

Experimental design

The major concern is why this three cultivars were selected and can be used to identify candidate genes involved in SCW biosynthesis, more specifically genes determining fiber strength. In addition, readers cannot get useful information from most of the main figures. Moreover, no experimental evidence shows that any of genes was indeed involved in SCW biosynthesis.

Validity of the findings

The major concern is why this three cultivars were selected and can be used to identify candidate genes involved in SCW biosynthesis, more specifically genes determining fiber strength. In addition, readers cannot get useful information from most of the main figures. Moreover, no experimental evidence shows that any of genes was indeed involved in SCW biosynthesis.

Additional comments

Other minor comments
1) Line25: What do HL, L and S stand for?
2) Line53-54: 15-25 days DPA, What do you mean days DPA?
3) Line57-58 Does cellulose have a hollow tube?
4) Line83 and less keratin, wax, inorganic matter, and protein that other plant cells. Please check the grammar
5) Line220-222 What do you mean?
6) Line239-249 Are there any difference between the three varieties? Be specific?
7) Line337-340 Do you mean 20-25 DPA represent transition stage?
8) Fig1-Fig8 I suggest that the authors focus on the main goal of this study which is to identify genes involved in fiber SCW thickening, and provide information related to this goal. I suppose that readers cannot get useful information from the figures.

Reviewer 2 ·

Basic reporting

The manuscript by Liu et al. employs RNA-seq to compare differentially expressed genes (DEGs) during the secondary cell thickening stage of cotton fiber development between upland cotton and sea island cotton. This study contributes valuable insights to cotton breeding. However, there are some minor concerns:

In lines 51-53, the number of stages for fiber development and the references may need verification, as there is uncertainty about whether there are four or five stages according to existing literature. Additionally, consider consulting the latest high-quality reviews on cotton fiber and updating the citations accordingly.

In line 194, there is an extra period after "S."

Lines 213-215: Consider moving this description to the Method section, as it may be more appropriate there.

Lines 324-331: Address the lack of quantitative data. When presenting real-time PCR results, it is advisable to describe the expression levels, such as indicating a percentage increase in the expression level of XXX. Additionally, for the heatmap, avoid using red and green colors together, as they may be visually challenging for readers.

In line 360, it should be GhMYB2. Please correct this typo.

Experimental design

The experiments conducted in this study demonstrate a well-designed approach. However, I recommend that the authors enhance the manuscript by incorporating a dedicated section on statistical analyses. This section should encompass discussions on statistical methods employed, emphasizing the importance of replication in the plant growth experiment. Additionally, provide detailed descriptions of the various analyses performed for phenotypic assessments or any other relevant aspects in the methodology. This inclusion will not only strengthen the experimental rigor but also provide readers with a clearer understanding of the statistical underpinnings supporting the study's findings.

Validity of the findings

no comment

---

## Round 0.2 · Major Revisions

Dear authors,

Post review of your manuscript, I have identified a number of issues which are of concern. Therefore the revision of your original submission requires some further work, as well as justification of some statements / information provided in the revision.

Please address my below concerns in a response letter upon resubmission.

Kind regards,

Andrew

My point-by-point comments are:

Please check English language standard throughout entire manuscript – numerous issues remain which require correction (many are highlighted in the annotated PDF)

Why are you reporting on analysis performed on samples collected in 2014 – 10 years ago? Surely these cultivars have been further refined in this time, and therefore, your analyses lack relevance – please justify

Sentence on lines 130-131 is a repeat of lines 126-127

Cottonseeds is not an acceptable term – it is cotton seeds – amend throughout.
Use cultivar in all instances – constant change of terminology when discussing cultivars studied is highly confusing to the reader

Line 134 – why are you using the term ‘transgenic’? This concerns me greatly, and indicates a lack of care when authoring the study. Why transgenic usage – please explain?

Lines 177-178 – why would you extract RNA from the leaves of cotton plants when you have studied gene expression in cotton fibres? This makes zero sense to makes, and remove any value to your RT-qPCR results. No relevance whatsoever to fibre development – please justify this decision

One base mismatch stated, then two mismatches stated – which is it? Further demonstration of a lack of care when authoring the manuscript, or a lack of experimental rigour.

The terms ‘expressions’ is not suitable in scientific writing – replace in all instances

The Discussion is of low quality and adds little to the study overall. No attempt to use the findings of others to support you own findings. This is concerning.

Why only analyse the expression via RT-qPCR of 11 genes, when so many DEGs were identified by sequencing? The RT-qPCR work fails to support your bioinformatic data.

Why no statistical analysis performed on RT-qPCR results? This is not acceptable, and must be done.

How can you explain a lack of correlation between sequencing and RT-qPCR expression trends for selected genes?

Reviewer 2 ·

Basic reporting

The authors have adressed all my concerns, and I recommend to publish the MS.

Experimental design

None.

Validity of the findings

None.

Additional comments

None

---

## Round 0.3 · Major Revisions

Dear authors,

Several textual issues remain throughout the text of the revised version of your manuscript - these must be addressed

Other major concerns are:

1. How can you identify more DEGs than are encoded for by the cotton genome? These obviously cannot represent unique protein coding gene transcripts. Therefore, what do these ~5,000 DEGs actually represent?

2. It is stated that two of the 11 genes assessed by RT-qPCR show no obvious expression trends. This statement is not supported by the expression data presented in Figure 6. This raises serious concerns that you are still failing to appreciate the significance of the gene expression data that you are presenting in the manuscript.

Considering that this will represent another major round of manuscript review, I need to state that if the requested changes are not made to an appropriate level in this round of revision, then I can no longer consider this study further in the PeerJ system. I am unwilling to perform additional rounds of review in an attempt to bring this study up to an acceptable standard.

Kind regards,
Andrew Eamens

---

## Round 0.4 · Minor Revisions

Dear authors,

Thank you for your efforts in greatly improving your manuscript.

For further final improvement please again thoroughly check you revised manuscript to ensure all textural issues have been addressed. I append a PDF with some suggestions

Kind regards,
Andrew

---

## Round 0.5 · accepted · Accept

Dear authors,

Thank you kindly for continuing to make the requested changes that I have made through the additional rounds of review of your original submission.

It is my opinion that your manuscript is now at a standard that can be accepted for publication in PeerJ. So again, thank you for your continued work through the multiple rounds of review, and congratulations on having your manuscript approved for publication.

All the best,
Andrew Eamens